# The Microstructures, Mechanical Properties, and Deformation Mechanism of B2-Hardened NbTiAlZr-Based Refractory High-Entropy Alloys

**DOI:** 10.3390/ma16247592

**Published:** 2023-12-11

**Authors:** Guangquan Tang, Xu Shao, Jingyu Pang, Yu Ji, Aimin Wang, Jinguo Li, Haifeng Zhang, Hongwei Zhang

**Affiliations:** 1Shi-Changxu Innovation Center for Advanced Materials, Institute of Metal Research, Chinese Academy of Sciences, Shenyang 110016, Chinajgli@imr.ac.cn (J.L.); 2School of Metallurgy, Northeastern University, Shenyang 110819, China

**Keywords:** refractory high-entropy alloys, B2 phase, specific yield strength, strain softening, deformation microstructure

## Abstract

The NbTiAlZrHfTaMoW refractory high-entropy alloy (RHEA) system with the structure of the B2 matrix (antiphase domains) and antiphase domain boundaries was firstly developed. We conducted the mechanical properties of the RHEAs at 298 K, 1023 K, 1123 K, and 1223 K, as well as typical deformation characteristics. The RHEAs with low density (7.41~7.51 g/cm^3^) have excellent compressive-specific yield strength ($\sigma_{\mathrm{YS}}$/ρ) at 1023 K (~131 MPa·cm^3^/g) and 1123 K (~104.2 MPa·cm^3^/g), respectively, which are far superior to most typical RHEAs. And, they still keep appropriate plastic deformability at room temperature (ε > 0.35). The superior specific yield strengths are mainly attributed to the solid solution strengthening induced by the Zr element. The formation of the dislocation slip bands with [111](10$\bar{1}$) and [111](11$\bar{2}$) directions and their interaction provide considerable plastic deformation capability. Meanwhile, dynamic recrystallization and dislocation annihilation accelerate the continuous softening after yielding at 1123 K.

## 1. Introduction

With the development of high-temperature metallic materials, Ni-based superalloys have been unable to meet the requirements of aerospace engines. It is urgent to find a new high-performance alloy to replace the Ni-based superalloy [1]. Refractory high-entropy alloys (RHEAs) based on refractory elements with superior high-temperature mechanical properties have recently attracted intensive research attention, which are regarded as promising candidates for high-temperature structural materials [2,3,4]. However, their inferior plasticity along with high ductile–brittle transition temperature and high density seriously limits their aerospace applications, where the requirements are mainly focus on the deformability and high specific strength of alloys.

Research efforts have been attempting to overcoming these shortcomings via component and structure regulation [5,6]. In order to improve plasticity at room temperature (RT), a new refractory high-entropy alloy (RHEA), HfNbTaTiZr [7], with excellent RT compression plasticity (ε > 50%), has been developed. However, its specific yield strength (SYS) at high temperatures is undesirability (such as <60 MPa·cm^3^/g at 1073 K). For better SYS, Han et al. [5] and Wang et al. [8] added Ti or V to NbMoTaW and NbMoTa RHEAs, respectively. Unfortunately, their SYSs are still lower than 90 MPa·cm^3^/g at 1073 K. Hence, the harmonization of strength, plasticity, and density is an intractable problem in RHEAs. Meanwhile, the strain softening phenomenon in RHEAs at high temperatures is ubiquitous, but the quantitative analysis of the softening mechanism is still lacking [9,10,11,12].

For the two crucial issues mentioned above, in this work, we realized the structural transition from disorder to order of a plastic NbTi-based RHEA via Al-Zr regulation. Meanwhile, high-temperature solid solution strengthening and grain boundary (GB) strengthening were also introduced into the RHEA by adding appropriate amounts of Hf, Ta, Mo, and W. On this basis, the fresh Nb_42_Ti_25_Al_15_Zr_5_Hf_5_Ta_5_Mo_2_W_1_ (NbTiAl-1), Nb_39_Ti_23_Al_15_Zr_10_Hf_5_Ta_5_Mo_2_W_1_ (NbTiAl-2), and Nb_37_Ti_20_Al_15_Zr_15_Hf_5_Ta_5_Mo_2_W_1_ (NbTiAl-3) RHEAs with the B2 matrix and antiphase domain boundaries (APBs) are firstly developed. The RHEAs exhibit excellent SYS at high temperatures, maintaining the top level in the field of plastic RHEAs. In addition, the deformation mechanism and strain softening characteristic at RT and 1123 K are also discussed in detail.

## 2. Materials and Methods

The RHEAs (the purity of elements was higher than 99.9 wt%) were prepared by vacuum melting in a furnace with a Ti-gettered argon atmosphere. The Zr-Ti-Al and Nb-Hf-Ta-Mo-W intermediate alloys were prepared, respectively in order to avoid the volatility of Al and Ti. Then, they were blended for the final button ingots, which were remelted at least six times to ensure chemical homogeneity. Compression samples with sizes of Φ5 mm × 7.5 mm and Φ8 mm × 12 mm were obtained directly from the button ingots. The 298 K compression tests (Φ5 mm × 7.5 mm) at RT were conducted on an Instron 5582 machine with a strain rate 0.001/s. The 1123 K compression tests (Φ8 mm × 12 mm) were conducted on a GLEEBLE 3800 machine with the same rate. Microstructure and composition analyses were performed with a scanning electron microscope (SEM; XL30-FEG, FEI, Eindhoven, Holland). In this work, both SEM and electron backscatter diffraction (EBSD) samples were, firstly, mechanically ground with 1200 to 2000 grit SiC sandpapers, followed by mechanical polishing. The mechanically polished surfaces were ultrasonically cleaned with alcohol and then electrolytically polished with an electrolyte of 6% HClO_4_ + 35% CH_3_(CH_2_)_3_OH + 59% CH_3_OH at about −30 °C. Phase identification was conducted using an X-ray diffractometer (XRD; Rigaku D/max-2500PC, Tokyo, Japan) and transmission electron microscope (TEM; Talos F200X, ThermoFisher, Waltham, MA, USA). The as-cast TEM samples, slices cut from the bulk samples, were mechanically ground to a thickness of ~70 μm. Disks with a diameter of 3 mm punched out from the thin foils were further ground to ~40 μm and finally thinned using a Gatan Model 695 after dimpling [13]. For TEM samples after deformation, thin specimens cut from the core of the column sample were mechanically ground to a thickness of approximately ~70 μm. The subsequent thinning process was consistent with the as-cast TEM samples.

## 3. Results and Discussion

The XRD results of the as-cast NbTiAl-1, NbTiAl-2, and NbTiAl-3 RHEAs are shown in Figure 1. All three alloys have a BCC structure. The XRD pattern of the NbTiAl-3 shows the presence of weakly ordered B2 phase structural diffraction peaks at about 26°. As shown in the enlarged illustration in Figure 1, the (110) diffraction peaks are obviously shifted to the left with the increase in Zr elements, indicating a significant increase in the lattice constants, which are 0.3302 nm for the NbTiAl-1, 0.3310 nm for the NbTiAl-2, and 0.3317 nm for the NbTiAl-3, respectively. Combined with the XRD results, Zr is the element with the largest atomic size in this alloy system, which results in the largest solid solution effect and thus a significant increase in the lattice constant. Meanwhile, the density values of NbTiAl-1, NbTiAl-2, and NbTiAl-3 are 7.51 g/cm^3^, 7.44 g/cm^3^, and 7.41 g/cm^3^, respectively.

As shown in Figure 2a, the as-cast structure of NbTiAl-1 is equiaxial dendritic segregation. NbTiAl-2 and NbTiAl-3 are dendritic structures, as shown in Figure 2b,c, which are mainly due to the rapid solidification process of elemental segregation. The EDS results shown in Figure 2d reveal that, due to the differences in the melting points of the constituent elements, the high-melting-point elements Nb, Ta, Mo, and W firstly solidify in the dendrite region (DR), and then drive the lower-melting-point Zr and Al elements into the inter-dendrite region (ID), which promotes the formation of the dendritic structure, whereas the distribution of the Ti and Hf elements is relatively uniform. This dendritic structure induced by the difference in elemental melting point is highly consistent with the as-cast microstructure of previous RHEAs [14,15].

Figure 3 shows the dark-field TEM (DF-TEM) images with the superlattice spots of (100) of as-cast NbTiAl-1, NbTiAl-2, and NbTiAl-3, which indicates that the as-cast structure of the RHEAs is composed of an ordered B2 matrix and APBs [16,17,18]. The presence of an ordered B2 structure is again verified by the superlattice spots in the electron diffraction pattern (EDP) with the [001] zone axis. This phenomenon is consistent with the result that the structure factor ratio $\frac{\left|F_{\left(100\right)}\right|}{\left|F_{\left(200\right)}\right|}$ is small due to a counterbalancing of heavy and lighter elements on the different lattice sites according to the site occupation factors $SOF_{i}^{LS}$ in MoCrTiAl [18]. From the DF-TEM results, the volume fraction of B2 phase keeps increasing with the increase in Zr element, i.e., the density of APBs decreases significantly because Al-Zr partitioning can promote the formation of the B2 structure during the rapid solidification process. This partitioning trend is consistent with previous studies [19,20]. It should be noted that the effect of the density of APBs on the yield strength of the RHEAs is limited because the APBs had no notable influence on the motion and spatial distribution of dislocations, as demonstrated in our previous work and that of others [21,22,23].

Figure 4 shows the compression true stress–strain curves of the NbTiAl-1, NbTiAl-2, and NbTiAl-3 at different temperatures. The yield strengths of the three RHEAs at 298 K are 1198.1 ± 7.5 MPa, 1333.1 ± 7.2 MPa, and 1418.1 ± 7.4 MPa, respectively. The yield strength of the RHEAs increases significantly with increasing Zr content. It indicates that the alloy has an obvious solid solution strengthening effect after the addition of Zr [17,24]. Similarly, the yield strengths of the RHEAs at 1023 K and 1123 K shown in Figure 4b,c maintain that trend, proving that the solid solution strengthening induced by the Zr element are still effective in that temperature range. The 1223 K compression results in Figure 4d show that the yield strengths of NbTiAl-2 and NbTiAl-3 are similar, indicating that the solid solution strengthening induced by the addition of Zr elements are weakening. The obvious serration behavior of NbTiAl-2 from dynamic strain aging (DSA), shown in Figure 4b, is caused by the tearing-off and re-pining of alternating dislocations [25,26].

In order to assess the effect of solid solution strengthening in the RHEAs, a power-law function was used to plot the relationship between the yield strength increment Δσ of NbTiAl-1, 2, and 3 RHEAs and the concentration of Zr in the B2 matrix (C*_Zr_*, Table 1): Δσ~$C_{Zr}^{n}$. A good linear fit between Δσ and C*_Zr_* was obtained for NbTiAl-1, 2, and 3 RHEAs at 298 K, 1023 K, and 1123 K when *n* = 0.25, as shown in Figure 5. Therefore, the increase in strength of NbTiAl-1, 2, and 3 RHEAs with increasing Zr content at 298 K, 1023 K, and 1123 K is doubtlessly caused by the solid solution strengthening effect. There is no longer a linear relationship between Δσ~$C_{Zr}^{0.25}$ at 1223 K, which proves that the solid solution strengthening is significantly reduced at this temperature, which is also consistent with the trend that the solid solution strengthening effect decreases with increasing temperature [27]. More interestingly, the strain softening of the RHEAs becomes more and more pronounced with the increase in Zr content in the range of 1023 K–1223 K. The specific strain softening mechanism will be analyzed below. The compressive yield strength (σ_YS_) and SYS (σ_YS_/ρ) of the three RHEAs at different temperatures are shown in Table 2.

The SYS values and strain of the RHEAs at RT are compared with other representative RHEAs, such as NbMoTa [28], AlNbTiV [24], ZrTiHfV_0.5_Nb_0.5_, Zr_2.0_TiHfVNb_2.0_ and ZrTiHfNb_0.5_Ta_0.5_ [29], Al_1.5_NbTa_0.5_Ti_1.5_Zr_0.5_ and AlMo_0.5_NbTa_0.5_TiZr [30], HfNbTaTiZr, HfMoTaTiZr, HfMoNbTiZr and HfMoNbTaTi [31], NbMoTaW and NbMoTaWV [4], CrNbTiVZr [32,33], and HfMo_0.5_NbTaTiZr [34]. The results show that Al_1.5_NbTa_0.5_Ti_1.5_Zr_0.5_, CrNbTiVZr, and AlMo_0.5_NbTa_0.5_TiZr RHEAs have high SYSs, but their plasticity is inferior, which seriously affects the application of these RHEAs as structural materials. Although the SYSs of NbTiAl-1, 2, and 3 RHEAs at RT are in the middle of the range of representative RHEAs, all of them have superior plasticity compared with most of the RHEAs. Namely, a good combination of the SYS and the compression plasticity is realized in these RHEAs. A comparison of the SYSs of NbTiAl-1, 2, and 3 RHEAs with other representative RHEAs at different temperatures is shown in Figure 6b, such as NbMoTaW and VNbMoTaW [4], TaNbHfZrTi [7], NbTaTiV [35], CrNbTiZr and CrNbTiVZr [32], Zr_2.0_TiHfVNb_2.0_, [29], and Ti_35_Al_15_V_20_Nb_20_Mo_10_ [36]. The results show that the SYSs of NbTiAl-1, 2, and 3 RHEAs at 298 K are only lower than that of two RHEAs, CrNbTiZr and CrNbTiVZr. However, the SYSs of the NbTiAl-1, 2, and 3 RHEAs are higher than that of other RHEA systems, and the SYS of the NbTiAl-3 is the highest in the temperature range of 973 K~1173 K. It is obvious that the NbTiAl-3 has a substantially higher SYS ($\sigma_{\mathrm{YS}}$/ρ) from 298 K (191.3 MPa·cm^3^/g) to 1123 K (112.7 MPa·cm^3^/g) compared with other typical RHEAs. The superior mechanical property is mainly attributed to the solid solution strengthening induced by the Zr element in the B2 matrix. Meanwhile, the critical shear stress for the slip of dislocations in ordered crystalline structures is much higher than that in their solid solution counterparts [17].

Serious plastic deformation occurs near the GB after the fracture of the NbTiAl-3, and a large number of slip bands appear. The slip bands have two orientations, represented as slip band 1 and slip band 2 in Figure 7a, respectively, with an orientation difference of about 26° between them. The two-beam condition BF-TEM image with **g** = (01$\bar{1}$) shows two typical dislocation slip systems ([111](10$\bar{1}$) and [111](11$\bar{2}$)), which are consistent with the orientation pile-up of slip bands shown in Figure 7a. There is an obvious dislocation pile-up in the slip bands while the two types of slip bands interact continually, thus providing considerable strain hardening capacity, as shown in Figure 4a. In particular, the appearance of {112} glide plane under RT deformation further provides the NbTiAl-3 with superior plastic deformation capability.

The electron backscatter diffraction (EBSD) results of the NbTiAl-3 after compression at 1123 K are shown in Figure 8. The compression axis (CA) is parallel to the vertical direction of Figure 8a. The grains are elongated along the horizontal direction, and the GBs slip significantly during compression, as shown in Figure 8a. Moreover, the geometrically necessary dislocation (GND) map shown in Figure 8b indicates that the dislocations are accumulated around the GBs, while the dislocation density is much lower inside the grains. Figure 8c shows that the dislocation density is 5.0 × 10^12^/m^2^ at the GBs, which is similar to the as-cast annealing state [37,38], whereas the dislocation densities around the GBs and inside the grains are 3.8 × 10^14^/m^2^ and 1.6 × 10^14^/m^2^, as shown in Figure 8d,e, respectively. It is precisely because of this heterogeneous deformation structure that it provides the necessary nucleation sites and driving forces for necklace structural dynamic recrystallization (DRX) at the GBs. Finally, the NbTiAl-3 forms the deformed structure that fine DRX grains are surrounded by high-density dislocations at the GBs after deformation, and the dislocation density gradually decreases, extending into the inside of the grains. Moreover, the overall dislocation density (10^12^/m^2^~10^14^/m^2^) of the sample is relative low [39], which proves that the dislocation annihilation occurs obviously during compression at 1123 K, resulting in the reduction in the dislocation strengthening effect. Hence, the DRX and dislocation annihilation accelerate the continuous strain softening after yielding.

The TEM results are also consistent with the EBSD analysis. Figure 9a shows the fine DRX grains along the initial GBs. Meanwhile, sub-grains with low-angle GBs are also detected, shown in Figure 9a, which indicates that the dynamic recovery (DRV) also occurs during compression. Accordingly, the dislocation density at the GBs is significantly reduced due to the consumption of DRX and DRV. Obvious dislocation tangles and high-density dislocation debris are detected surrounding the DRX and DRV regions, as shown in Figure 9b. Despite the fact that the NbTiAl-3 RHEA has superior SYS conferred by the existence of the ordered B2 matrix, DRX and DRV effectively reduce the stored elastic energy by consuming dislocations along the GBs. Meanwhile, dislocation cross-slip/climb and annihilation accelerated by thermal actuation are apparent inside the grains at 1123 K, resulting in continuous strain softening [9,10]. Figure 9c shows a schematic diagram of the microstructure of recrystallization and dislocation after compression at 1123 K.

## 4. Conclusions

Overall, three novel RHEAs, Nb_42_Ti_25_Al_15_Zr_5_Hf_5_Ta_5_Mo_2_W_1_ (NbTiAl-1), Nb_39_Ti_23_Al_15_Zr_10_Hf_5_Ta_5_Mo_2_W_1_ (NbTiAl-2), and Nb_37_Ti_20_Al_15_Zr_15_Hf_5_Ta_5_Mo_2_W_1_ (NbTiAl-3), with the B2 matrix and APBs were developed. The microstructure, RT, and high-temperature mechanical properties of the RHEAs in the as-cast state were systematically analyzed, and the deformation mechanism was also studied by taking the NbTiAl-3 RHEA as an example, with the following specific conclusions:(1)The RHEAs exhibit an excellent combination of high yield strength, low density (7.41~7.51 g/cm^3^), and large plasticity (ε > 0.35) at RT. The SYSs of the NbTiAl-3 RHEA are ~131 MPa·cm^3^/g at 1023 K and ~104.2 MPa·cm^3^/g at 1123 K, respectively, which are far superior to most typical RHEAs. These excellent mechanical properties are mainly attributed to the solid solution strengthening induced by the Zr element.(2)The deformation of the NbTiAl-3 RHEA at 298 K is dominated by the planar slip of dislocations, resulting in two dislocation slip bands oriented in the [111](10$\bar{1}$) and [111](11$\bar{2}$) directions. Dislocation pile-up in the slip bands and the two types of slip bands interact continually, providing considerable plastic deformation capability.(3)The DRX formed at GBs and low dislocation density inside the grains induced via dislocation annihilation accelerate continuous strain softening after yielding during the 1123 K compression of the NbTiAl-3 RHEA.

## Figures and Tables

**Figure 1 materials-16-07592-f001:**
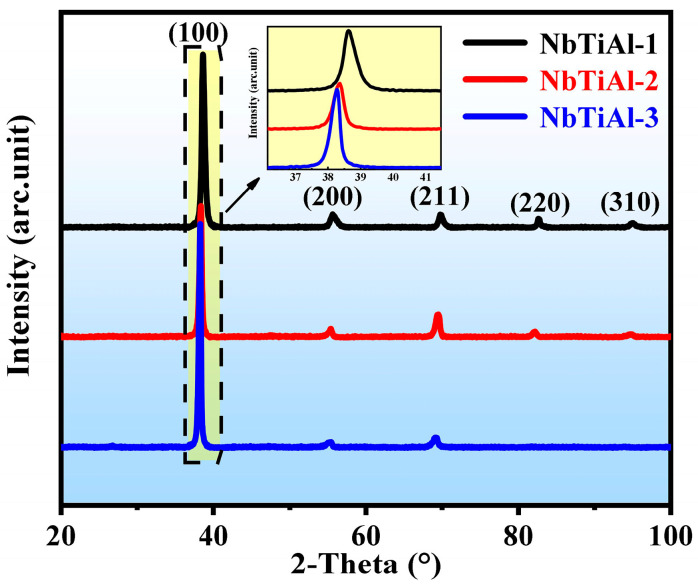
XRD pattern of as-cast NbTiAl-1, NbTiAl-2, and NbTiAl-3 RHEAs.

**Figure 2 materials-16-07592-f002:**
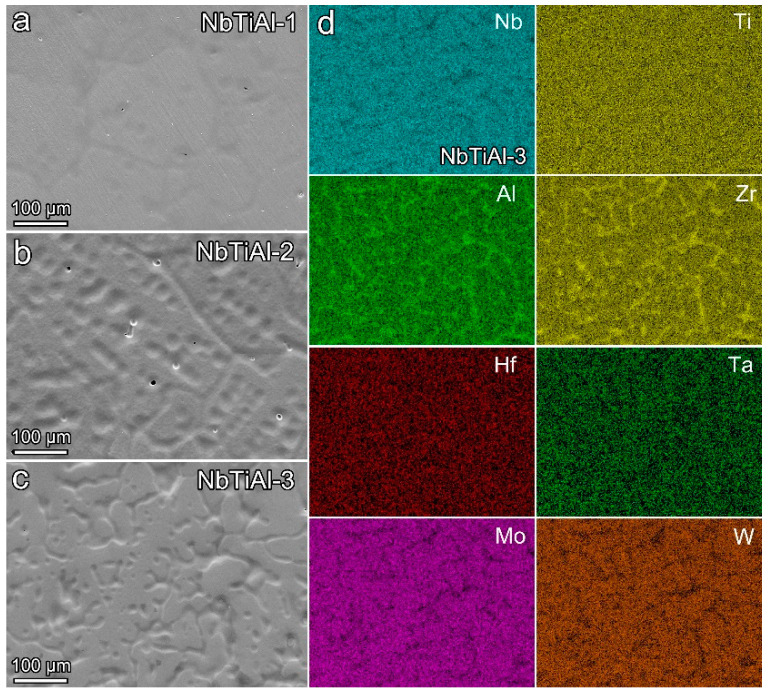
SEM images of as-cast (**a**) NbTiAl-1, (**b**) NbTiAl-2, (**c**) NbTiAl-3 RHEAs, (**d**) EDS maps of as-cast NbTiAl-3 RHEA.

**Figure 3 materials-16-07592-f003:**
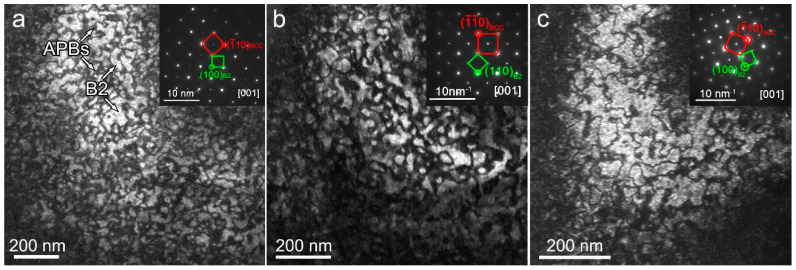
DF-TEM images with the superlattice spots of (100) of as-cast (**a**) NbTiAl-1, (**b**) NbTiAl-2, (**c**) NbTiAl-3 RHEAs. The insets in (**a**–**c**) are selected-area electron diffraction (SAED) patterns with [001] zone axis.

**Figure 4 materials-16-07592-f004:**
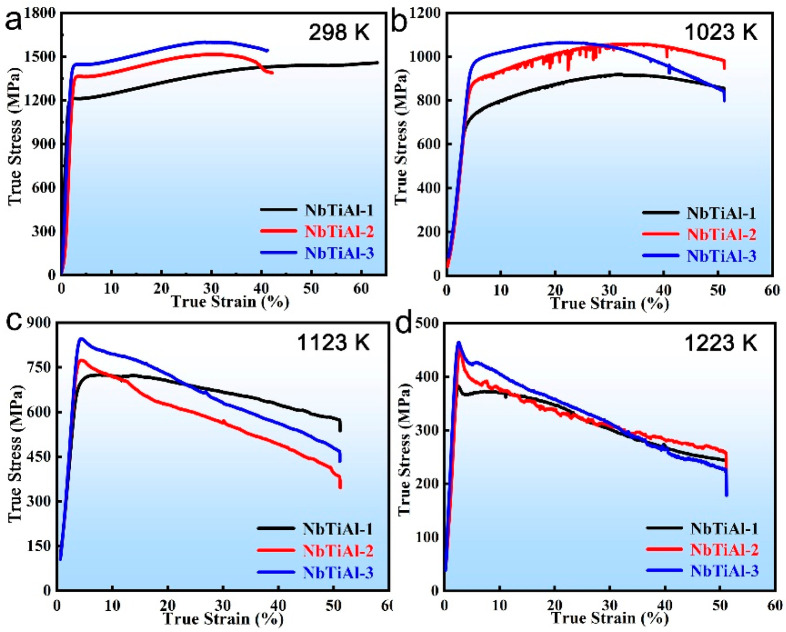
Compressive true stress–strain curves of as-cast NbTiAl-1, NbTiAl-2, and NbTiAl-3 RHEAs at (**a**) 298 K, (**b**) 1023 K, (**c**) 1123 K, and (**d**) 1223 K.

**Figure 5 materials-16-07592-f005:**
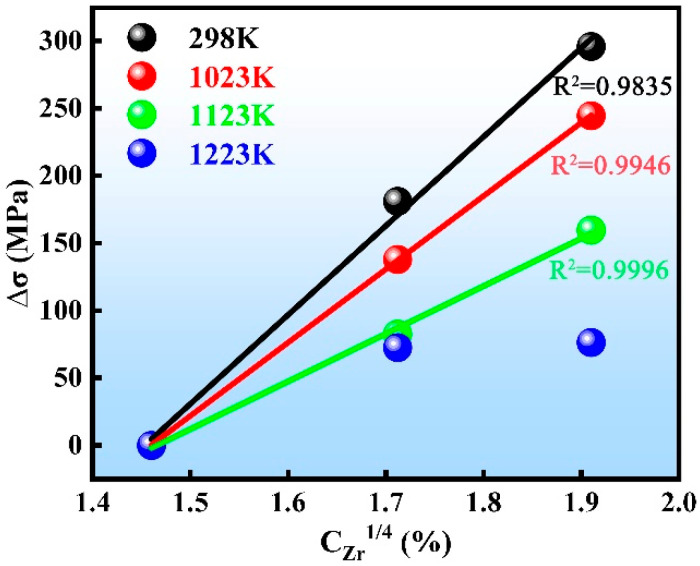
The increment in the yield strength, Δσ, at T = 298 K, 1023 K, 1123 K, and 1223 K as a function of the Zr concentration, C_Zr_, in the matrix phase of the RHEAs.

**Figure 6 materials-16-07592-f006:**
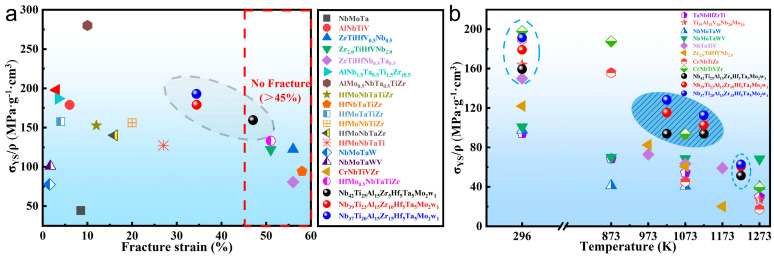
(**a**) The SYS ($\sigma_{\mathrm{YS}}$/ρ)-fracture strain of as-cast NbTiAl-1, NbTiAl-2, and NbTiAl-3 RHEAs and other representative RHEAs at RT, (**b**) the SYS ($\sigma_{\mathrm{YS}}$/ρ)-temperature of as-cast NbTiAl-1, NbTiAl-2, and NbTiAl-3 RHEAs and other representative RHEAs [4,7,24,28,29,30,31,32,33,34,35,36].

**Figure 7 materials-16-07592-f007:**
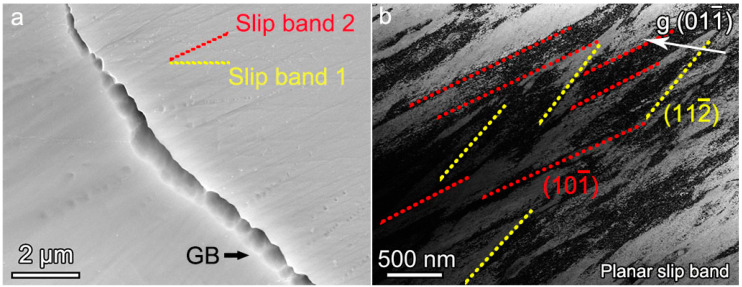
(**a**) SEM image showing two independent slip traces and (**b**) two-beam condition BF-TEM image with **g** = (01$\bar{1}$) showing two typical dislocation slip systems ([111](10$\bar{1}$) and [111](11$\bar{2}$)) of as-cast NbTiAl-3 RHEA after fracture at 298 K.

**Figure 8 materials-16-07592-f008:**
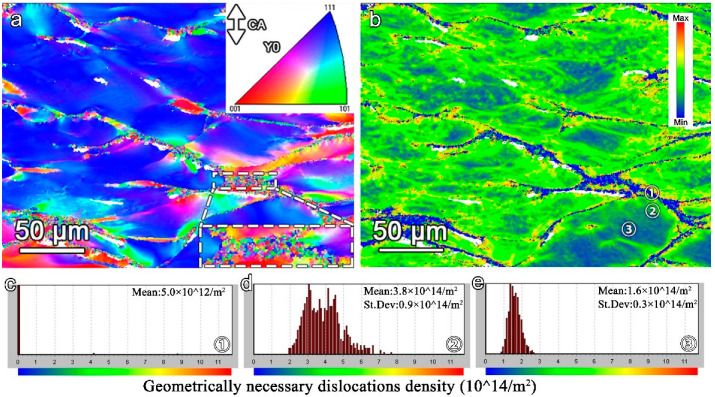
EBSD results after compression true strain 50% at 1123 K. (**a**) Inverse pole figure (IPF) maps. (**b**) GND maps. (**c**–**e**) GND values corresponding to the selected regions ①–③, at the GBs, nearby the GBs, and the grain interior, respectively, shown in (**b**).

**Figure 9 materials-16-07592-f009:**
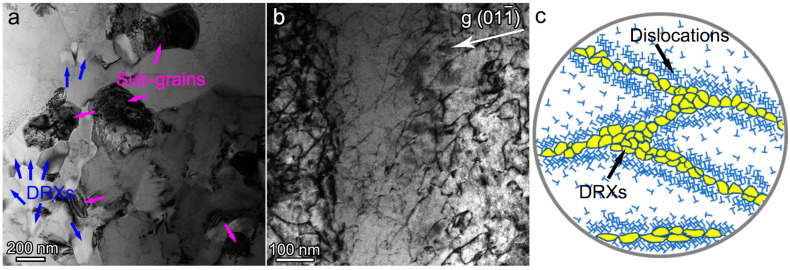
(**a**) BF-TEM image, (**b**) two-beam condition BF-TEM image with **g** = (00$\bar{1}$), and (**c**) deformation schematic diagram of as-cast NbTiAl-3 RHEA after 50% compression at 1123 K.

**Table 1 materials-16-07592-t001:** Chemical compositions of the as-cast NbTiAl-1, 2, and 3 RHEAs (at.%).

Nominal Composition	Nb	Ti	Al	Zr	Hf	Ta	Mo	W
Nb_42_Ti_25_Al_15_Zr_5_Hf_5_Ta_5_Mo_2_W_1_	41.2	25.7	14.3	4.6	5.2	6.0	1.4	1.6
Nb_39_Ti_23_Al_15_Zr_10_Hf_5_Ta_5_Mo_2_W_1_	40.4	23.7	13.6	8.6	5.1	5.8	1.1	1.7
Nb_37_Ti_20_Al_15_Zr_15_Hf_5_Ta_5_Mo_2_W_1_	37.7	20.3	14.1	13.5	5.3	6.2	1.8	1.0

**Table 2 materials-16-07592-t002:** Values of compression yield strength and SYS of as-cast NbTiAl-1, 2, and 3 RHEAs at different temperatures (Unit: MPa and MPa·g^−1^·cm^3^).

	Temperature (K)	298	1023	1123	1223
NbTiAl-1	σYS	1198.1 ± 7.5	705.6 ± 12.8	705.5 ± 12.5	386.2 ± 9.2
	σYS/ρ	159.5	93.9	93.9	51.4
NbTiAl-2	σYS	1333.1 ± 7.2	857.9 ± 13.1	764.6 ± 12.7	451.8 ± 11
	σYS/ρ	179.1	115.3	102.7	60.7
NbTiAl-3	σYS	1418.1 ± 7.4	951.8 ± 11.5	835.5 ± 13	466.3 ± 11.4
	σYS/ρ	191.3	128.4	112.7	62.9

## Data Availability

Data is contained within the article.

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
