# Peer review of "The Microstructures, Mechanical Properties, and Deformation Mechanism of B2-Hardened NbTiAlZr-Based Refractory High-Entropy Alloys"

_materials, 2023, doi:10.3390/ma16247592_

Round 1

Reviewer 1 Report

Comments and Suggestions for Authors

Deformation behavior and mechanism were clarified for high entropy B2 single phase. High strength at 1123 K is very promissing to design new high-temperature materials. The manuscript will be accepted after minor revision.

1. Show the reason why good elongation at RT.

2. Show the experimantal procedure more precisely.

    for example, 

   sample preparation for TEM, TEM microscopy name, company

   how much strain was applied for EBSD observation?

Author Response

Reviewer 1#: Deformation behavior and mechanism were clarified for high entropy B2 single phase. High strength at 1123 K is very promissing to design new high-temperature materials. The manuscript will be accepted after minor revision.

  1. Show the reason why good elongation at RT.

Reply 1: Thank you very much for your valuable advice. We have added the discussion about the good Elongation at RT in the manuscript as follows:

“The two-beam condition BF-TEM image with g = (01-1) showing two typical dislocation slip systems ([111](10-1) and [111](11-2)), which are consistent with the orientation pile-up of slip bands shown in Fig. 7(a). There is obvious dislocation pile-up in the slip bands while the two types of slip bands interact continually, thus providing considerable strain hardening capacity, as shown in Fig. 4(a). In particular, the appearance of {112} glide plane under RT deformation further provides the NbTiAl-3 with superior plastic deformation capability.”

  1. Show the experimantal procedure more precisely. for example, sample preparation for TEM, TEM microscopy name, company. how much strain was applied for EBSD observation?

Reply 2: Thank you very much for your valuable advice. We have added the sample preparation for TEM, TEM microscopy name, company and the strain for EBSD results in the manuscript as follows:

“Phase identification was conducted using an X-ray diffractometer (XRD; Rigaku D/max-2500PC) with Cu-Kα radiation in the 2θ range of 20°-100° and transmission electron microscope (TEM; Talos F200X, ThermoFisher). The as-cast TEM samples, slices cut from the bulk samples were mechanically ground to a thickness of ~70 μm. Disks with a diameter of 3 mm punched out from the thin foils were further ground to ~40 μm and finally thinned using a Gatan Model 695 after dimpling [13]. For TEM samples after deformation, thin specimens cut from the core of the column sample were mechanically ground to a thickness of approximately ~70 μm. The subsequent thinning process was consistent with the as-cast TEM samples.

Fig. 8. EBSD results after compression true strain 50% at 1123 K. (a) Inverse pole figure (IPF) maps. (b) GND maps. (c-e) GND values corresponding to the selected regions 1-3, at the GBs, nearby the GBs, grain interior, respectively, shown in (b).

Reviewer 2 Report

Comments and Suggestions for Authors

This article discusses an investigation into the mechanical behavior and microstructural properties of a NbTiAlZrHfTaMoW refractory high entropy alloy that underwent compression testing at temperatures ranging from ambient to 1273 K. They report that the alloy exhibited excellent compressive specific yield strength that was superior to other refractory high entropy alloys. An interesting article although I have a few comments.

1. In the abstract, experimental conditions such as temperature should be completely listed.

2. The word fresh does not seem to be the correct word in the abstract, please replace it with something more fitting.

3. A lot of details are missing from the Materials and methods section, namely:

   - What are the purity of the elements used to fabricate the samples?

   - How were samples prepared for the scanning electron microscope characterization?

   - Energy of the Cu X-ray?

   - How were focused ion beam samples prepared for the transmission electron microscopy characterization?

4. There appear to be some serrations in the stress vs. strain data featured in Figure 4, please give a broader discussion on this behavior. Some references below can aid in this process:

- Journal of Alloys and Compounds 930 (2023); Acta Mater. 242 (2023) 118445; Mater. Sci. Eng. A-Struct. Mater. Prop. Microstruct. Process. 850 (2022).

5. Stress is spelled incorrectly in the y-axis of Figure 4.

6. Please provide R2 values for the fitted lines in Fig. 5.

7. Please provide references for the listed materials in Figure 6. 

Comments on the Quality of English Language

There are some moderate grammatical errors in the text. For example, in the abstract, the sentence that beings with "And They" is incorrectly written.

Author Response

Reviewer #2: This article discusses an investigation into the mechanical behavior and microstructural properties of a NbTiAlZrHfTaMoW refractory high entropy alloy that underwent compression testing at temperatures ranging from ambient to 1273 K. They report that the alloy exhibited excellent compressive specific yield strength that was superior to other refractory high entropy alloys. An interesting article although I have a few comments.

  1. In the abstract, experimental conditions such as temperature should be completely listed.

Reply 1: Thank you very much for your valuable advice. We have added the experimental conditions in the abstract as follows:

“We conducted the mechanical properties of the RHEAs at 298 K, 1023 K, 1123 K, and 1223 K, as well as typical deformation characteristics.”

  1. The word fresh does not seem to be the correct word in the abstract, please replace it with something more fitting.

Reply 2: Thank you very much for your valuable advice. We have deleted the word “fresh” in the abstract as follows:

“The NbTiAlZrHfTaMoW refractory high entropy alloys (RHEAs) system with the structure of B2 matrix (antiphase domains) and antiphase domain boundaries was firstly developed.”

  1. A lot of details are missing from the Materials and methods section, namely:

   - What are the purity of the elements used to fabricate the samples?

Reply: Thank you very much for your valuable question. We have added the purity of the elements in the manuscript as follows:

“The RHEAs (the purity of elements higher than 99.9 wt %) were prepared by vacuum melting furnacein a Ti-gettered argon atmosphere.”

   - How were samples prepared for the scanning electron microscope characterization?

Reply: Thank you very much for your valuable question. We have added the samples prepared process for the SEM and EBSD in the manuscript as follows:

“In this work, both SEM and electron backscatter diffraction (EBSD) samples, were firstly mechanically ground with 1200 to 2000 grit SiC sandpapers, followed by mechanical polishing. The mechanically polished surfaces were ultrasonically cleaned with alcohol and then electrolytically polished with an electrolyte of 6% HClO4 + 35% CH3(CH2)3OH + 59% CH3OH at about -30 ℃.”

   - Energy of the Cu X-ray?

Reply: Thank you very much for your valuable question. We apologize for our sloppy presentation. And we have modified the description in the manuscript as follows:

“Phase identification was conducted using an X-ray diffractometer (XRD; Rigaku D/max-2500PC) and transmission electron microscope (TEM; Talos F200X, ThermoFisher).”

   - How were focused ion beam samples prepared for the transmission electron microscopy characterization?

Reply: Thanks for your question. In our present work, all the TEM samples were obtained by the traditional preparation method, and the detailed preparation process we have added in the manuscript. The FIB technique for preparing TEM samples has been widely used, but due to the small thin area, the observation and calibration for dislocations will be limited.

“The as-cast TEM samples, slices cut from the bulk samples were mechanically ground to a thickness of ~70 μm. Disks with a diameter of 3 mm punched out from the thin foils were further ground to ~40 μm and finally thinned using a Gatan Model 695 after dimpling [13]. For TEM samples after deformation, thin specimens cut from the core of the column sample were mechanically ground to a thickness of approximately ~70 μm. The subsequent thinning process was consistent with the as-cast TEM samples.”

  1. There appear to be some serrations in the stress vs. strain data featured in Figure 4, please give a broader discussion on this behavior. Some references below can aid in this process:- Journal of Alloys and Compounds 930 (2023); Acta Mater. 242 (2023) 118445; Mater. Sci. Eng. A-Struct. Mater. Prop. Microstruct. Process. 850 (2022).

Reply 4: Thank you very much for your valuable advice. Due to the complexity of serrations, the mechanism of its formation is not discussed in detail in this paper. And follow the authors’ advice, we briefly outlined the reasons for the formation of s serrations and cited the corresponding literatures. It should be noted that two of the literatures provided by the reviewer were not available to us due to insufficient information. If necessary, we ask the reviewer to provide detailed information so that we can cite them. We discuss the serrations behavior as follows:

“The obvious serration behavior of NbTiAl-2 from dynamic strain aging (DSA), shown in Fig. 4(b), is caused by tearing-off and re-pining of dislocations alternating [25-27].”

  1. Stress is spelled incorrectly in the y-axis of Figure 4.

Reply 5: Thank you very much for your kindly reminding. We apologize for the incorrect spelling! We have modified the Fig. 4 as follows:

“”

  1. Please provide R2 values for the fitted lines in Fig. 5.

Reply 6: Thank you very much for your valuable advice. We have added the R2 values for the fitted lines in Fig. 5 as follow:

“”

  1. Please provide references for the listed materials in Figure 6.

Reply 7: Thank you very much for your valuable advice. We apologize for our negligence. We have added the references for the listed materials in Fig. 6, as follows”

Fig. 6. (a) The SYS (/)-fracture strain of as-cast NbTiAl-1, NbTiAl-2, NbTiAl-3 RHEAs and other representative RHEAs at RT, (b) the SYS (/)-temperature of as-cast NbTiAl-1, NbTiAl-2, NbTiAl-3 RHEAs and other representative RHEAs [4, 7, 24, 26-33, 36].”